# NeuroBack: Improving CDCL SAT Solving using Graph Neural Networks

**Wenxi Wang, Yang Hu, Mohit Tiwari, Sarfraz Khurshid, Ken McMillan, Risto Miikkulainen**
The University of Texas at Austin
Austin, TX 78712, USA
`wenxiw@utexas.edu,huyang@utexas.edu,tiwari@austin.utexas.edu,`
`khurshid@utexas.edu,kenmcm@cs.utexas.edu,risto@cs.utexas.edu`

## Abstract

Propositional satisfiability (SAT) is an NP-complete problem that impacts many research fields, such as planning, verification, and security. Mainstream modern SAT solvers are based on the Conflict-Driven Clause Learning (CDCL) algorithm. Recent work aimed to enhance CDCL SAT solvers using Graph Neural Networks (GNNs). However, so far this approach either has not made solving more effective, or required substantial GPU resources for frequent online model inferences. Aiming to make GNN improvements practical, this paper proposes an approach called NeuroBack, which builds on two insights: (1) predicting phases (i.e., values) of variables appearing in the majority (or even all) of the satisfying assignments are essential for CDCL SAT solving, and (2) it is sufficient to query the neural model *only once* for the predictions before the SAT solving starts. Once trained, the offline model inference allows NeuroBack to execute exclusively on the CPU, removing its reliance on GPU resources. To train NeuroBack, a new dataset called DataBack containing 120,286 data samples is created. NeuroBack is implemented as an enhancement to a state-of-the-art SAT solver called `Kissat`. As a result, it allowed `Kissat` to solve up to 5.2% and 7.4% more problems on two recent SAT competition problem sets, SATCOMP-2022 and SATCOMP-2023, respectively. NeuroBack therefore shows how machine learning can be harnessed to improve SAT solving in an effective and practical manner.

## 1 Introduction

Propositional satisfiability (SAT) solvers are designed to check the satisfiability of a given propositional logic formula. SAT solvers have advanced significantly in recent years, which has fueled their applications in a wide range of domains. Mainstream modern SAT solvers are based on the CDCL algorithm Marques-Silva & Sakallah (2003). It mainly relies on two kinds of variable related heuristics: 1) the variable branching heuristic Moskewicz et al. (2001); Liang et al. (2016), for deciding which variables are the best to branch on; 2) the phase selection heuristic Pipatsrisawat & Darwiche (2007); Biere & Fleury (2020); Fleury & Heisinger (2020), for deciding which phase (i.e., value) a variable should have.

Recently, Graph Neural Networks (GNNs) have been proposed to improve CDCL SAT solvers by enhancing these two heuristics. However, this approach has not usually made solvers more effective Jaszczur et al. (2020); Kurin et al. (2019); Han (2020). One exception is NeuroCore Selsam & Bjørner (2019) which aims to improve the variable branching heuristic in CDCL SAT solvers by predicting the variables involved in the unsatisfiable (unsat) core (i.e., a subset of the SAT formula that remains unsat). It performs frequent online model inferences to adjust its unsat core prediction with dynamic information extracted from the SAT solving process. As a result, NeuroCore helps two classic CDCL SAT solvers called `MiniSat` and `Glucose` solve more problems in the main track of SATCOMP-2018 Heule et al. (2018).

However, frequent online inferences cause NeuroCore computationally intensive during its application. The demand on GPU resources escalate, especially when deployed in parallel, a prevalent way for utilizing SAT solvers to tackle complex problems Hamadi et al. (2010); Schreiber & Sanders (2021); Martins et al. (2012). Obviously, lacking GPU resources would become the major bottleneck of NeuroCore's parallel performance. Note that, the setup of NeuroCore's experiments called for network access to 20 GPUs distributed over five machines to support 400 parallel NeuroCore runs.

We propose NeuroBack, a novel approach to make CDCL SAT solving more effective and avoid frequent online model inferences, thus making the GNN approach more practical. The main idea of NeuroBack is to make *offline* model inference, i.e., prior to the solving process, to obtain instructive static information for improving CDCL SAT solving. Once trained, the offline model inference allows NeuroBack to execute solely on the CPU, thereby making it completely independent of GPU resources. In particular, NeuroBack seeks to refine the phase selection heuristics in CDCL solvers by leveraging offline neural predictions on variable phases appearing in the majority (or even all) of the satisfying assignments.

The offline predictions on such phase information are based on the generalization of backbone variables, which are variables whose phases remain consistent across all satisfying assignments. Recent work Biere et al. (2021); Al-Yahya et al. (2022) has shown that backbone variables are crucial for enhancing CDCL SAT solving. Choosing the correct phase for a backbone variable prevents conflicts, while an incorrect choice inevitably leads to backtracking in the search. Moreover, predicting the correct phases of non-backbone variables appearing in the majority of satisfying assignments is also important, because such phases prevent backtracking with high probabilities. Our conjecture is that the knowledge learned from predicting the phases of backbone variables can be transferred to predicting the phases of non-backbone variables exhibited in the majority of satisfying assignments. Therefore, NeuroBack applies a GNN model, trained solely on predicting the phases of backbone variables, to predict the phases of variables appearing in the majority (or even all) of the satisfying assignments. We refer to such variables as *important variables* for the rest of this paper.

NeuroBack converts the SAT formula with diverse scales into a compact and more learnable graph representation, turning the problem of predicting variable phases into a binary node classification problem. To make the GNN model both compact and robust, NeuroBack employs a novel Graph Transformer architecture with light-weight self-attention mechanisms. To train the model with supervised learning, a *balanced* dataset called DataBack containing 120,286 labeled formulas with diversity was created from five different sources: CNFgen Lauria et al. (2017), SATLIB Hoos & Stützle (2000), model counting competitions from 2020 to 2022 MCC (2020; 2021; 2022), and main and random tracks in SAT competitions from 2004 to 2021.

To evaluate the effectiveness of our approach, NeuroBack is incorporated into a state-of-the-art CDCL SAT solver called `Kissat` Biere & Fleury (2022), resulting in a new solver called `NeuroBack-Kissat`. The experimental results on all SAT problems from SATCOMP-2022 SAT (2022) and SATCOMP-2023 SAT (2023) show that NeuroBack allows `Kissat` to solve up to 5.2% and 7.4% more problems, respectively. The experiments thus demonstrate that NeuroBack is a practical neural approach to improving CDCL SAT solvers. The contributions of our paper are:

**1. Approach.** To our knowledge, NeuroBack presents the first practical neural approach to make the CDCL SAT solving more effective, without requiring any GPU resource during its application.

**2. Dataset.** A new dataset DataBack containing 120,286 data samples is created for backbone phase classification. DataBack is publicly available at `https://huggingface.co/datasets/neuroback/DataBack`

**3. Implementation.** NeuroBack is incorporated into a state-of-the-art SAT solver, `Kissat`. The source code of NeuroBack model and `NeuroBack-Kissat` is publicly available at `https://github.com/wenxiwang/neuroback`.

## 2 BACKGROUND

This section introduces the SAT problem, CDCL algorithm, phase selection heuristics in CDCL solvers, and basics of GNN and Graph Transformer.

**Preliminaries of SAT** In SAT, a propositional logic formula $\phi$ is usually encoded in Conjunctive Normal Form (CNF), which is a conjunction ($\wedge$) of *clauses*. Each clause is a disjunction ($\vee$) of *literals*. A literal is either a variable $v$, or its complement $\neg v$. Each variable can be assigned a logical phase, 1 (true) or 0 (false). A CNF formula has a satisfying assignment if and only if every clause has at least one true literal. For example, a CNF formula $\phi = (v_1 \vee \neg v_2) \wedge (v_2 \vee v_3) \wedge v_2$ consists of three clauses $v_1 \vee \neg v_2$, $v_2 \vee v_3$ and $v_2$; four literals $v_1$, $\neg v_2$, $v_2$ and $v_3$; and three variables $v_1$, $v_2$, and $v_3$. One satisfying assignment of $\phi$ is $v_1 = 1$, $v_2 = 1$, $v_3 = 0$. The goal of a SAT solver is to check if a formula $\phi$ is sat or unsat. A *complete* solver either outputs a satisfying assignment for $\phi$, or proves that no such assignment exists.

The backbone of a sat formula is the set of literals that are true in all its satisfying assignments. Thus backbone variables are the variables whose phases remain consistent across all satisfying assignments. In the given CNF formula $\phi$, there are two backbone variables, $v_1$ and $v_2$, both maintaining a phase of 1 in all satisfying assignments of $\phi$.

**CDCL Algorithm**    CDCL makes SAT solvers efficient in practice and is one of the main reasons for the widespread use of SAT applications. The general idea of CDCL algorithm is as follows (see Marques-Silva et al. (2021) for details). First, it picks a variable on which to branch and decides a phase to assign to it with heuristics. It then conducts a Boolean propagation based on the decision. In the propagation, if a conflict occurs (i.e., at least one clause is mapped to 0), it performs a conflict analysis; otherwise, it makes a new decision on another selected variable. In the conflict analysis, CDCL first analyzes the decisions and propagations to investigate the reason for the conflict, then extracts the most relevant wrong decisions, undoes them, and adds the reason to its memory as a learned lesson (encoded in a clause called *learned clause*) in order to avoid making the same mistake in the future. After conflict analysis, the solver *backtracks* to the earliest decision level where the conflict could be resolved and continues the search from there. The process continues until all variables are assigned a phase (sat), or until it learns the empty clause (unsat).

**Phase Selection Heuristics**    As indicated above, CDCL SAT solving mainly relies on two kinds of variable related heuristics: variable branching heuristics and phase selection heuristics, which are orthogonal to each other. Phase Saving Pipatsrisawat & Darwiche (2007) is a prevalent phase selection heuristic in modern CDCL solvers. It returns a variable's last assigned polarity, either through decision or propagation. This heuristic addresses the issue of solvers forgetting prior valid assignments due to non-chronological backtracking. Rephasing Biere & Fleury (2020); Fleury & Heisinger (2020) is more recent phase selection heuristic, proposed to reset or modify saved phases to diversify the exploration of the search space. The state-of-the-art CDCL solver, `Kissat` Biere & Fleury (2022), incorporates the latest advancements in phase saving and rephasing heuristics.

**GNN**    GNNs Wu et al. (2020); Zhou et al. (2020) are a family of neural network architectures that operate on graphs Gori et al. (2005); Scarselli et al. (2008); Gilmer et al. (2017); Battaglia et al. (2018). Typical GNNs follow a recursive neighborhood aggregation scheme called message passing Gilmer et al. (2017). Formally, the input of a GNN is a graph defined as a tuple $G = (V, E, W, H)$, where $V$ denotes the set of nodes; $E \subseteq V \times V$ denotes the set of edges; $W = \{W_{u,v} | (u, v) \in E\}$ contains the feature vector $W_{u,v}$ of each edge $(u, v)$; and $H = \{H_v | v \in V\}$ contains the feature vector $H_v$ of each node $v$. A GNN maps each node to a vector-space embedding by updating the feature vector of the node iteratively based on its neighbors. For each iteration, a message-passing layer $\mathcal{L}$ takes a graph $G = (V, E, W, H)$ as an input and outputs a graph $G' = (V, E, W, H')$ with updated node feature vectors, i.e., $G' = \mathcal{L}(G)$. Classic GNN models Gilmer et al. (2017); Kipf & Welling (2016); Hamilton et al. (2017) usually stack several message-passing layers to realize iterative updating. Prior work on utilizing GNNs to improve CDCL SAT solving will be reviewed in the next section.

**Graph Transformer Architecture**    Transformer Vaswani et al. (2017) is a family of neural network architectures for processing sequential data (e.g., text or image), which has recently won great success in nature language processing and computer vision. Central to the transformer is the self-attention mechanism, which calculates attention scores among elements in a sequence, thereby allowing each element to focus on other relevant elements in the sequence for capturing long-range dependencies effectively. Recent research Min et al. (2022); Wu et al. (2021); Mialon et al. (2021); Rong et al. (2020) has shown that combining the transformer with GNN results in competitive performance for graph and node classification tasks, forming the Graph Transformer architecture. Notably, `GraphTrans` Wu et al. (2021) is a representative model which utilizes a GNN subnet consisting of multiple GNN layers for local structure encoding, followed by a Transformer subnet with global self-attention to capture global dependencies, and finally incorporates a Feed Forward Network (FFN) for classification. The GNN model design in NeuroBack is inspired by `GraphTrans`.

## 3   RELATED WORK

This section presents related work on identifying the backbone and machine learning techniques for improving CDCL SAT solving.

Figure 1: Overview of NeuroBack. First, the input CNF formula is converted into a compact and more learnable graph representation. A trained GNN model is then applied once on the graph before SAT solving begins for phase selection. The SAT solver utilizes phase information in the resulting labeled graph as an initialization to guide its solving process. Thus, with the offline process of making instructive phase predictions, NeuroBack makes the solving more effective and practical.

**Backbone for CDCL Solvers.** Janota proved that identifying the backbone is co-NP complete Janota (2010). Furthermore, Kilby et al. demonstrated that even approximating the backbone is generally intractable Kilby et al. (2005). Wu Wu (2017) applies a logistic regression model to predict the phase of backbone variables to improve a classic CDCL SAT solver called MiniSat Eén & Sörensson (2003). Although the approach correctly predicts the phases of 78% backbone variables, it fails to make improvements over MiniSat in solving time. In addition, recent works Biere et al. (2021); Al-Yahya et al. (2022) have been focusing on enhancing CDCL SAT solving by employing heuristic search to partially compute the backbone during the solving process. In contrast, NeuroBack applies GNN to predict the backbone in an offline manner to improve CDCL solving.

**Machine Learning for CDCL Solvers.** Recently, several approaches have been developed to utilize GNNs to facilitate CDCL SAT solving. NeuroSAT Selsam et al. (2018) was the first such framework adapting a neural model into an end-to-end SAT solver, which was not intended as a complete SAT solver. Others Jaszczur et al. (2020); Davis et al. (1962); Kurin et al. (2019); Han (2020); Audemard & Simon (2018); Zhang & Zhang (2021) aim to provide SAT solvers with better branching or phase selection heuristics. These approaches either reduce the number of solving iterations or enhance the solving effectiveness on selected small-scale problems with up to a few thousand variables. However, they do not provide obvious improvements in solving effectiveness for large-scale problems.

In contrast, NeuroCore Selsam & Bjørner (2019), the most closely related approach to this paper, aims to make the solving more effective especially for large-scale problems as in SAT competitions. It enhances the branching heuristic for CDCL using supervised learning to map unsat problems to unsat core variables (i.e., the variables involved in the unsat core). Based on the dynamically learned clauses during the solving process, NeuroCore performs frequent online model inferences to tune the predictions. However, this online inference is computationally demanding. NeuroBack is distinct from NeuroCore in two main aspects. One, while NeuroCore is designed to refine the branching heuristic in CDCL SAT solvers, NeuroBack is invented to enhance their phase selection heuristics. Two, while NeuroCore extracts dynamic unsat core information from unsat formulas through online model inferences, NeuroBack captures static backbone information from sat formulas using offline model inference. Details of NeuroBack are introduced in the following section.

## 4 NEUROBACK

### 4.1 OVERVIEW

In order to reduce the computational cost of the online model inference and to make CDCL SAT solving more effective, NeuroBack employs offline model predictions on variable phases to enhance the phase selection heuristics in CDCL solvers. Figure 1 shows the overview of NeuroBack. First, it converts the input CNF formula into a compact and more learnable graph representation. Then, a well-designed GNN model trained to predict the phases of backbone variables, is applied on the converted graph representation to infer the phases of important variables. The model inference is performed only once before the SAT solving process. The resulting offline prediction is applied as an initialization for the SAT solving process. Finally, the enhanced SAT solver outputs the satisfiability of the input CNF formula. Key components of NeuroBack including the graph representation of CNF formulas, the GNN-based phase selection, and the phase prediction application in SAT solvers, are illustrated in the subsections below.

### 4.2 GRAPH REPRESENTATION FOR CNF FORMULAS

As in recent work Kurin et al. (2020); Yolcu & Póczos (2019), a SAT formula is represented using a more compact undirected bipartite graph than the one adopted in NeuroCore. Two node types

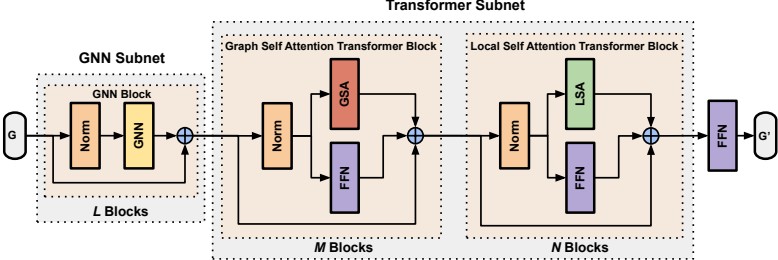

Figure 3: The architecture of NeuroBack model, consisting of three main components: a GNN subnet with $L$ stacked GNN blocks, a transformer subnet with $M$ GSA transformer blocks and $N$ LSA transformer blocks, and a FFN layer for node classification.

represent the variables and clauses, respectively. Each edge connects a variable node to a clause node, representing that the clause contains the variable. Two edge types represent two polarities of a variable appearing in the clause, i.e., the variable itself and its complement. Although the representation is compact, its diameter might be substantial for large-scale SAT formulas, which could result in insufficient message passing during the learning process.

To mitigate this issue, we introduce a *meta node* for each connected component in the graph, with meta edges connecting the meta node to all clause nodes in the component. With the added meta nodes and edges, every variable node not appearing in the same clauses can reach each other through their corresponding clause nodes and the meta node, thereby making the diameter at most four. Figure 2 shows an example of our graph representation for the CNF formula $(v_1 \lor v_2) \land (v_2 \lor v_3) \land (v_3 \lor v_4)$. It includes one meta node, four variable nodes and three clause nodes $c_1, c_2, c_3$, representing clauses $v_1 \lor v_2$, $v_2 \lor v_3$ and $v_3 \lor v_4$, respectively. Without the added meta node and edges, the longest path in the graph runs from variable node $v_1$ to variable node $v_4$, making the diameter six. However, by introducing the meta node and edges, the diameter is reduced to four.

Formally, for a graph representation $G = (V, E, W, H)$ of a CNF formula, the edge feature $W_{u,v}$ of each edge $(u, v)$ is initialized by its edge type with the value $0$ representing the meta edge, the value $1$ representing positive polarity, and $-1$ negative polarity; the node feature $H_v$ of each node $v$ is initialized by its node type with $0$ representing a meta node, $1$ representing a variable node, and $-1$ representing a clause node.

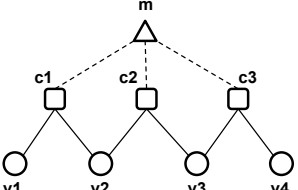

Figure 2: An example graph representation of the CNF formula $(v_1 \lor v_2) \land (v_2 \lor v_3) \land (v_3 \lor v_4)$. A meta node $m$ is added along with meta edges (represented by dashed lines) connecting to all clause nodes in the connected component, reducing the graph diameter from six to four.

### 4.3 GNN-BASED PHASE PREDICTION

Given that the phases of backbone variables remain consistent across all satisfying assignments, selecting the correct phase for a backbone variable prevents backtracking. Conversely, an incorrect choice inevitably leads to a conflict. Moreover, the proportion of backbone variables is typically significant. For instance, during our data collection from five notable sources (i.e., CNFgen, SATLIB, model counting competitions, SATCOMP random tracks, and SATCOMP main tracks), backbone variables constitute an average of 27% of the total variables. Therefore, accurately identifying the phases of the backbone variables is crucial for efficiently solving a SAT formula. Furthermore, identifying the phases of the non-backbone variables appearing in the majority of satisfying assignments is also important. Because such phases could prevent backtracking with high probabilities.

Inspired by transfer learning Zhuang et al. (2020), we first train a GNN model to predict the phases of backbone variables, and then leverage the trained model to predict the phases of important variables. Our key insight is that the knowledge learned from predicting the phases of backbone variables can provide a valuable guidance on predicting the phases of non-backbone variables exhibiting in the majority of satisfying assignments. With the converted graph representation, predicting the phases of all variables is a binary node classification problem for GNN, where $1$ and $0$ represent the positive and negative phases, respectively. To extract the phases of important variables, we introduce a threshold $\alpha$ (set to $0.9$ in our current implementation). If the predicted probability is greater than $\alpha$ or less than $1 - \alpha$, it is classified as the positive or negative phase of an important variable, respectively. The following subsections introduce the design, implementation, and training of our GNN model.

### 4.3.1 GNN MODEL DESIGN

The sizes of converted graphs representing practical SAT formulas (usually with millions of variables and clauses) are typically substantial. To enable effective training within the constraints of limited GPU memory, it is essential for our model to be both compact and robust. Our GNN model design is inspired by the robust graph transformer architecture, `GraphTrans` Wu et al. (2021). However, in our particular SAT application context, `GraphTrans` exhibits two limitations, both arising from the global self-attention mechanism within its transformer subnet. First, the mechanism does not explicitly integrate the topological graph structure information when determining attention scores. However, such information is essential in characterizing a SAT formula. Second, the global self-attention mechanism computes attention scores for all possible node pairs, leading to *quadratic* memory complexity with respect to the number of nodes in the graph. This is obviously infeasible for tackling the large-scale SAT formulas in our task.

To overcome the limitations, we introduce a novel transformer subnet that both distinctly harnesses topological structure information and significantly enhances memory efficiency. It combines Graph Self-Attention (GSA) and Local Self-Attention (LSA), replacing the original transformer's global self-attention. Instead of computing attention scores for all node pairs as global self-attention, GSA calculates attention scores solely for directly connected node pairs, leveraging information of edges and edge weights. This not only explicitly incorporates the topological structure information of the graph, but also reduces the memory complexity to linear in terms of the number of edges in the graph. Additionally, to further reduce the memory complexity, LSA segments each node embedding into multiple node patches and computes attention scores for each pair of node patches. The results in a linear memory complexity in terms of the number of nodes in the graph.

Figure 3 illustrates the design of our GNN model architecture. It consists of three main components: a GNN subnet with $L$ stacked GNN blocks, a transformer subnet with $M$ GSA transformer blocks and $N$ LSA transformer blocks, and a FFN layer for node classification. Within each GNN block, a GNN layer is preceded by a normalization layer, with a skip connection bridging the two. The transformer block is crafted to accelerate training on a significant collection of large-scale graphs. Inspired by the recent vision transformer architecture, `ViT-22B` Padlewski & Djolonga (2023), each transformer block integrates a normalization layer, succeeded by both an FFN layer and a GSA/LSA layer that operate concurrently to optimize training efficiency.

### 4.3.2 IMPLEMENTATION

The current implementation of NeuroBack model utilizes `GINConv` Fey & Lenssen (2023b); Xu et al. (2018) to build the GNN layer, for its proficiency in distinguishing non-isomorphic graph structures. However, `GINConv` lacks the capability to encode edge weight information. To address this, we employ three `GINConv` layers, each corresponding to a distinct edge weight in our graph representation. Each `GINConv` layer exclusively performs message passing for edges with its corresponding weight. The node embeddings from these three `GINConv` layers are finally aggregated as the output of the GNN layer.

`GATConv` layers Fey & Lenssen (2023a); Veličković et al. (2017) is utilized to built the GSA transformer block. The patch encoder in the `ViT` transformer Dosovitskiy et al. (2020) is applied to construct the LSA transformer block. `LayerNorm` Ba et al. (2016) is employed as our normalization layer. To avoid potential over-smoothing issues, as instructed in Chen et al. (2020), the number of blocks in the GNN subnet is set to the maximum diameter of the graph representation (i.e., $L = 4$). To ensure the accuracy of our model, while taking into account our limited GPU memory, the number of both GSA and LSA blocks are set to three (i.e., $M = 3$ and $N = 3$). Additionally, FFNs within the transformer blocks contain no hidden layers, while the final FFN utilized for node classification is structured to include one hidden layer. The model is implemented using PyTorch Paszke et al. (2019) and PyTorch Geometric Fey & Lenssen (2019).

### 4.3.3 MODEL PRE-TRAINING AND FINE-TUNING

The NeuroBack model undergoes a two-stage training process. Initially, it is pre-trained on an extensive and diverse dataset gathered from various sources. This pre-training equips it with the fundamental knowledge to classify the backbone variable phases across a broad spectrum of CNF formulas. Subsequently, this pre-trained model is refined or fine-tuned on a smaller, domain-specific dataset. This fine-tuning process enhances the model's proficiency in classifying backbone variable

| DataBack-PT | CNFgen | SATLIB | MCCOMP | SATCOMP (random) | Overall | DataBack-FT | SATCOMP (main) |
|---|---|---|---|---|---|---|---|
| **# CNF** | 21,718 | 80,306 | 11,560 | 4,876 | 118,460 | **# CNF** | 1,826 |
| **# Var** | 779 | 114 | 16,299 | 25,310 | 2,852 | **# Var** | 206,470 |
| **# Cla** | 281,888 | 529 | 63,501 | 88,978 | 61,898 | **# Cla** | 1,218,519 |
| **# Backbone Var** | 280 (36%) | 58 (51%) | 8,587 (53%) | 1,960 (8%) | 1,009 (35%) | **# Backbone Var** | 48,266 (23%) |

Table 1: Details of DataBack. For each source, we summarize the number of CNF formulas; the average number of variables and clauses characterizing the size of each CNF formula; and the average number of backbone variables, with their proportion in total variables (indicated in brackets). Overall, for each dataset, we summarize the total number of CNF formulas and the corresponding average counts regarding all formulas in the set.

phases within a specific category of CNF formulas. Details about the pre-training and fine-tuning datasets are introduced in Section 5.

Binary Cross Entropy (BCE) loss is adopted as our loss function. Besides, AdamW optimizer Loshchilov & Hutter (2017) with a learning rate of $10^{-4}$ is applied for model pre-training and fine-tuning. The number of epoch is set to 40 for pre-training, and 60 for fine-tuning. It took 48 hours in total to accomplish both the pre-training and fine-tuning on our commodity computer.

### 4.4 APPLYING PHASE PREDICTIONS IN SAT

The goal is to leverage phase predictions derived from the GNN model to enhance the phase selection heuristics within CDCL SAT solvers. While numerous ways exist to integrate these neural predictions into CDCL SAT solvers, the most straightforward and generic approach is to initialize the phases of the corresponding variables in CDCL SAT solvers based on the predicted phases. In this paper, we adopted the state-of-the-art solver `Kissat` Biere & Fleury (2022) to support the phase initialization with NeuroBack predictions. The resulting implementation is called `NeuroBack-Kissat`.

## 5 DATABACK

We created a new dataset, DataBack, comprising sat CNF formulas labeled with phases of backbone variables, for pre-training (PT) and fine-tuning (FT) the NeuroBack model. Accordingly, there are two subsets in DataBack: the pre-training set, DataBack-PT, and the fine-tuning set, DataBack-FT. To get the backbone label, the very recent state-of-the-art backbone extractor called CadiBack Biere et al. (2023) is utilized. Given that the fine-tuning formulas are typically more challenging to solve than the pre-training ones, label collection timeouts are set as 1,000 seconds for the pre-training formulas and 5,000 seconds for the fine-tuning formulas. DataBack includes formulas solved within the time limit with at least one backbone variable.

We observe that there exists a significant label imbalance in both DataBack-PT and DataBack-FT. To tackle this, for each labeled formula, we create a dual formula by negating all backbone variables in the original formula. The labels of the dual formula are the negated phases of the backbone variables in the original formula. Formally, for a formula $f$ with $n$ backbone variables $b_1, \ldots, b_n$, let $\mathcal{L}_f : \{b_1, \ldots, b_n\} \to \{1, 0\}$ denote the mapping of each backbone variable to its phase. The dual formula $f'$ is obtained from $f$ by negating each backbone variable: $f' = f[b_1 \mapsto \neg b_1, \ldots, b_n \mapsto \neg b_n]$. The dual $f'$ is still satisfiable and retains the same backbone variables as $f$, but with the opposite phases $\mathcal{L}_{f'}(b_i) = \neg \mathcal{L}_f(b_i), i \in \{1, \ldots, n\}$. For the given CNF formula example in Section 2 $\phi = (v_1 \vee \neg v_2) \wedge (v_2 \vee v_3) \wedge v_2$, having $v_1$ and $v_2$ as its backbone variables with phases $\{v_1, v_2\} \to \{1\}$, the dual formula is $\phi' = (\neg v_1 \vee v_2) \wedge (\neg v_2 \vee v_3) \wedge \neg v_2$, still having $v_1$ and $v_2$ as the backbone variables but with opposite phases $\{v_1, v_2\} \to \{0\}$. This data augmentation strategy doubles the size of DataBack with a perfect balance in positive and negative backbone labels. In the rest of the paper, DataBack-PT and DataBack-FT refer to the augmented, balanced datasets.

**DataBack-PT** The CNF formulas in DataBack-PT are sourced from four origins: 1) CNFgen Lauria et al. (2017), a recent CNF generator renowned for crafting CNF formulas that feature in proof complexity literature; 2) SATLIB Hoos & Stützle (2000), an online benchmark library housing well-known benchmark instances, serving as a foundational challenge for the SAT community; 3) model counting competitions from 2020 to 2022 Fichte et al. (2021); MCC (2021; 2022), including wide range of CNF formulas for model counters to count their satisfying assignments, which provide a good source for backbone extraction; 4) Random tracks in SAT competitions from 2004 to 2021, offering numerous challenging random SAT problems.

| model \ metrics | precision | recall | $F_1$ | accuracy |
|---|---|---|---|---|
| pre-trained model | 0.903 | 0.766 | 0.829 | 0.751 |
| fine-tuned model | 0.941 | 0.914 | 0.928 | 0.887 |

Table 2: The performance on the validation set of both pre-trained and fine-tuned NeuroBack models for classifying the phases of backbone variables, in terms of precision, recall, F1 score and accuracy.

Table 1 shows the detailed statistics of the resulting dataset. CNFgen generated 10,859 labeled sat formulas (21,718 augmented formulas) grouped by five categories: random SAT, clique-coloring, graph coloring, counting principle, and parity principle. From SATLIB, 40,153 sat formulas with labels (80,306 augmented formulas) were collected, encoding seven kinds of problems: random SAT, graph coloring, planning, bounded model checking, Latin square, circuit fault analysis, and all-interval series. Model counting competitions and SAT competition random tracks contribute 5,780 (11,578 augmented) and 2,438 (4,876 augmented) labeled formulas, respectively. In total, 59,230 (118,460 augmented) formulas are included DataBack-PT.

The comparative size of formulas, as indicated by the average number of variables and clauses, reveals that formulas from model counting competitions and SAT competition random tracks are generally larger than those generated by CNFgen, which are in turn larger than those originating from SATLIB. In addition, formulas from model counting competitions and SATLIB exhibit relatively high average backbone proportions. These are followed by formulas from CNFgen, while formulas from the SAT competition's random tracks demonstrate the lowest backbone proportion. DataBack-PT thus contains a diverse set of formulas.

**DataBack-FT** Given that NeuroBack will be tested on large-scale SAT formulas from the main track of recent SAT competitions, the fine-tuning dataset, DataBack-FT, incorporates CNF formulas from the main track of earlier SAT competitions spanning from 2004 to 2021. As shown in Table 1, it contains 913 (1,826 augmented) labeled formulas. While DataBack-FT is considerably smaller in size compared to DataBack-PT, its individual formulas are distinctly larger than those in DataBack-PT.

## 6 EXPERIMENTS

**Platform** All experiments were run on an ordinary commodity computer with one NVIDIA GeForce RTX 3080 GPU (10GB memory), one AMD Ryzen Threadripper 3970X processor (64 logical cores), and 256GB RAM.

**Research Questions** The experiments aim to answer two research questions:
RQ1: *How accurately does the NeuroBack model classify the phases of backbone variables?*
RQ2: *How effective is the NeuroBack approach?*

**RQ1: NeuroBack Model Performance** The NeuroBack model was pre-trained on the entire DataBack-PT dataset, then fine-tuned on a random 90% of DataBack-FT samples, and evaluated on the remaining 10% as a validation set. Table 2 details the performance of both the pre-trained and fine-tuned models in classifying the phases of backbone variables. Notably, the pre-trained model achieved 75.1% accuracy in classifying backbone variables, with a precision exceeding 90% and a recall rate of 76.7%. Considering the distinct data sources of DataBack-PT and DataBack-FT, the results suggest that the pre-training enables the model to extract generalized knowledge about backbone phase prediction. Fine-tuning further augments model performance, with improvements ranging between 4% and 15% across all metrics, making precision, recall and F1 score all exceeding 90%. *In conclusion, NeuroBack model effectively learns to predict the phases of backbone variables through both pre-training and fine-tuning.*

**RQ2: NeuroBack Performance** To evaluate the solving effectiveness, we collect all 800 CNF formulas from the main track of SATCOMP-2022 SAT (2022) and SATCOMP-2023 SAT (2023) as our testing dataset. For our baseline solvers, we selected the default configuration of `Kissat`, named `Default-Kissat`, which simply sets the initial phase of each variable to true. We implemented an additional baseline solver, `Random-Kissat`, which randomly assigns the initial phase of each variable as either true or false. `NeuroBack-Kissat` and its baseline solvers, were applied to all 800 SAT problems in the testing dataset, with the standard solving time limit of 5,000 seconds. Each solver utilized up to 64 different processes in parallel on the dedicated 64-core machine. The model inference for each NeuroBack solver was conducted solely on the CPU, with a memory limit of 10GB to mitigate memory contention issues. Consequently, 308 problems from SATCOMP-2022 and

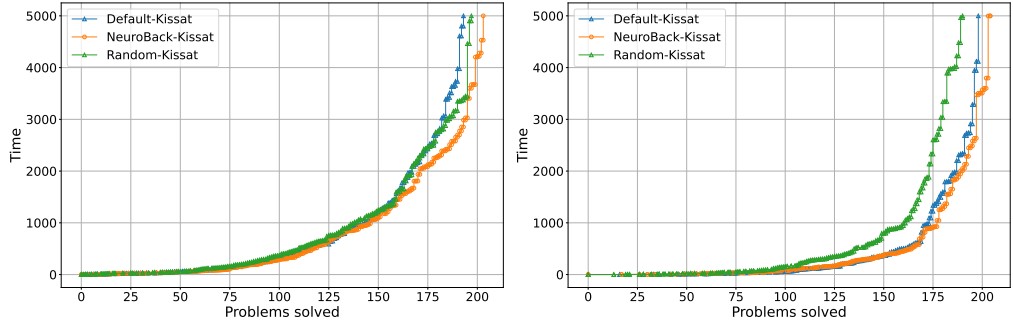

(a) Solving progress in SATCOMP-2022    (b) Solving progress in SATCOMP-2023

Figure 4: Progress of `Default-Kissat`, `Random-Kissat`, and `NeuroBack-Kissat` over time in solving problems (time in seconds) on SATCOMP-2022 (left) and SATCOMP-2023 (right), respectively. `NeuroBack-Kissat` outperforms the two baseline solvers on both testing sets.

353 problems from SATCOMP-2023 were successfully inferred. The CPU inference time for each of these problems ranged from 0.3 to 16.5 seconds, averaging at 1.7 seconds. Given that problems unsuccessfully inferred make the NeuroBack solver revert to its baseline, we exclude such problems from our evaluation of how NeuroBack contributes to SAT solving.

As a result, for 308 inferred problems in SATCOMP-2022, `Default-Kissat` and `Random-Kissat` solved 193 and 197 problems, respectively. In comparison, `NeuroBack-Kissat` solved 203 problems, representing an improvement of 5.2% over `Default-Kissat` and 3.0% over `Random-Kissat`, respectively. For 353 inferred problems in SATCOMP-2023, `Default-Kissat` and `Random-Kissat` successfully solved 198 and 190 problems, respectively. In contrast, `NeuroBack-Kissat` solved 204 problems, making improvements of 3.0% and 7.4% over `Default-Kissat` and `Random-Kissat`, respectively.

The cactus plot is a commonly used plot in the SAT community for demonstrating the solving, as shown in Fig. 4. `Random-Kissat` performs better than `Default-Kissat` in SATCOMP-2022, but worse in SATCOMP-2023. In contrast, `NeuroBack-Kissat` consistently outperforms both baseline solvers. Specifically, `NeuroBack-Kissat` outperforms `Default-Kissat` on 43 and 40 additional problems in SATCOMP-2022 and SATCOMP-2023, respectively, reducing solving time by 117 and 36 seconds per problem. Similarly, `NeuroBack-Kissat` outperforms `Random-Kissat` in SATCOMP-2022 and SATCOMP-2023 on 22 and 29 more problems, respectively, reducing solving time by 98 and 246 seconds per problem. For additional results, please refer to the Appendix section. *Overall, results suggest that the phase initialization provided by NeuroBack outperforms both the default and random phase initializations in `Kissat`, exhibiting enhanced proficiency in solving SATCOMP-2022 and SATCOMP-2023 problems.*

## 7 DISCUSSION AND FUTURE WORK

The current implementation of NeuroBack has two main aspects to improve. First, pre-training the GNN model requires a large amount of data. Once trained, however, the model is able to easily generalize to new problem categories, so the investment in putting together a large dataset is warranted. The dataset is publicly available to benefit future research. Second, this paper employs neural phase predictions solely as an initial setting for variable phases in SAT solvers. However, multiple methods exist to utilize these predictions in SAT, either in a static or dynamic manner, which are likely to yield further performance enhancements. For instance, phase predictions could be leveraged dynamically during rephasing to adjust or diversify the exploration of the search space.

## 8 CONCLUSION

This paper proposes a machine learning approach, NeuroBack, to make CDCL SAT solvers more effective without requiring any GPU resource during its application. The main idea is to make offline model inference on variable phases appearing in the majority of satisfying assignments, for enhancing the phase selection heuristic in CDCL solvers. Incorporated in the state-of-the-art SAT solver, `Kissat`, this approach significantly reduces the solving time and makes it possible to solve more instances in SATCOMP-2022 and SATCOMP-2023. NeuroBack is thus a promising approach to improving the SAT solvers through machine learning.

## 9 ACKNOWLEDGMENT

We would like to thank the anonymous reviewers for their valuable feedback. This work was supported by a grant from the Army Research Office accomplished under Cooperative Agreement Number W911NF-19-2-0333. The views and conclusions contained in this document are those of the authors and should not be interpreted as representing the official policies, either expressed or implied, of the Army Research Office or the U.S. Government. The U.S. Government is authorized to reproduce and distribute reprints for Government purposes notwithstanding any copyright notation herein. This work was also supported in part by ACE, one of the seven centers in JUMP 2.0, a Semiconductor Research Corporation (SRC) program sponsored by DARPA and by the Intel RARE center.

## 10 APPENDIX

### 10.1 ADDITIONAL EXPERIMENTAL RESULTS

The scatter plots is another commonly used plot in the SAT community for comparing the solving effectiveness of two solvers on each problem. Fig. 5 shows the scatter plots of `NeuroBack-Kissat` and its two baseline solvers, `Default-Kissat` and `Random-Kissat`. It is evident that more dots are present in the lower triangular area, indicating that there are more problems on which `NeuroBack-Kissat` outperforms both `Default-Kissat` and `Random-Kissat`. Specifically, `NeuroBack-Kissat` outperforms `Default-Kissat` on 43 and 40 additional problems in SATCOMP-2022 and SATCOMP-2023, respectively, reducing solving time by 117 and 36 seconds per problem. Similarly, `NeuroBack-Kissat` outperforms `Random-Kissat` in SATCOMP-2022 and SATCOMP-2023 on 22 and 29 more problems, respectively, leading to a reduction in solving time of 98 and 246 seconds per problem.

### 10.2 PERFORMANCE ON SOLVED SAT AND UNSAT PROBLEMS

Upon detailed analysis, for 661 problems from both SATCOMP-2022 and SATCOMP-2023 testing sets, there are 194 unsat problems and 216 sat problems that are solved by either `Default-Kissat` or `NeuroBack-Kissat`. For the 194 solved unsat problems, `NeuroBack-Kissat` outperformed `Default-Kissat` in 121 cases (62.4%) while `Default-Kissat` outperformed `NeuroBack-Kissat` in only 61 problems (31.4%). For the 216 solved sat problems, `NeuroBack-Kissat` outperformed `Default-Kissat` in 110 problems (50.9%), while `Default-Kissat` outperformed `NeuroBack-Kissat` in 87 problems (40.3%). While `NeuroBack-Kissat` showed a higher improvement rate in unsat problems compared to sat ones (62.4% vs 50.9%), the extent of improvement was more significant in sat problems. On average, `NeuroBack-Kissat` enhanced the performance of sat problems by 53.2%, compared to an average improvement of only 14.6% in unsat problems. These trends were similarly observed when comparing `NeuroBack-Kissat` with `Random-Kissat`.

The experimental results highlight two key aspects. First, they demonstrate that NeuroBack's predicted variable phases can enhance the efficiency in solving unsat problems. Our explanation is that NeuroBack's phase predictions can aid in directing the search towards the unsatisfiable part of the search space. While NeuroBack cannot satisfy all components of a given SAT problem, it may predict phases that satisfy certain components, thereby allowing the solver to concentrate on the unsat part. Furthermore, in modern SAT solvers such as `Default-Kissat` Biere & Fleury (2020), an assignment that falsifies the fewer clauses is often preferred in the searching loop, allowing the solver to specifically target the unsat portions of the clause set. Consequently, the phases predicted by NeuroBack can facilitate identifying an assignment that reduces clause falsification, thereby enhancing solving unsat problems.

Second, the experimental results also show that NeuroBack achieves a more pronounced improvement in solving sat problems than in solving unsat problems. This distinction stems from the inherent nature of these problems. In sat problems, a complete satisfying assignment exists, where each variable is assigned a phase that leads to a solution. Conversely, in unsat problems, only partial satisfying assignments exist, with phases assigned to just a subset of variables. Consequently, the phases predicted by NeuroBack have a generally greater impact in resolving sat problems. This is because, for these problems, the predicted phases can contribute directly to finding a satisfying assignment. In contrast, for unsat problems, the utility of predicted phases is somewhat restricted to identifying partial solutions or refining the search scope. This fundamental difference in the nature of

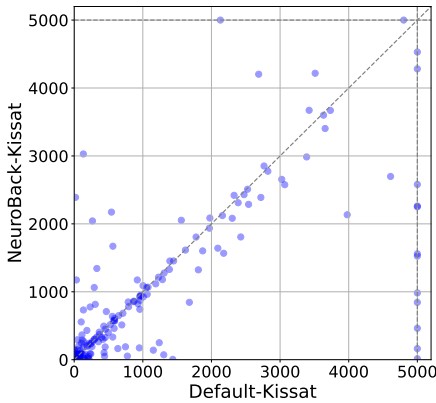

(a) Default-Kissat vs. NeuroBack-Kissat on SATCOMP-2022.

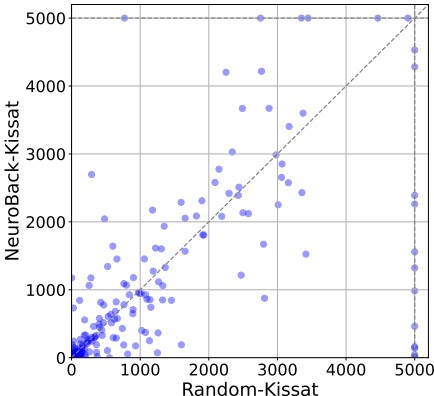

(b) Random-Kissat vs. NeuroBack-Kissat on SATCOMP-2022.

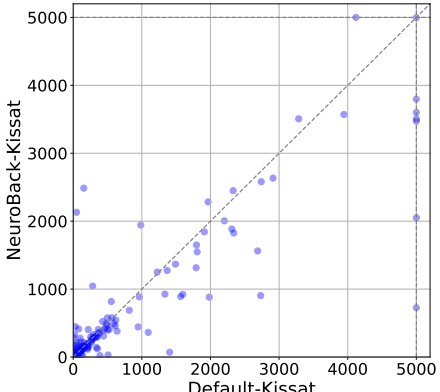

(c) Default-Kissat vs. NeuroBack-Kissat on SATCOMP-2023.

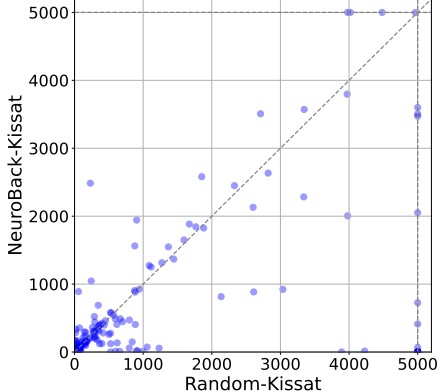

(d) Random-Kissat vs. NeuroBack-Kissat on SATCOMP-2023.

Figure 5: Time taken by `Default-Kissat` vs. `NeuroBack-Kissat` on SATCOMP-2022 (a), `Random-Kissat` vs. `NeuroBack-Kissat` on SATCOMP-2022 (b), `Default-Kissat` vs. `NeuroBack-Kissat` on SATCOMP-2023 (c), and `Random-Kissat` vs. `NeuroBack-Kissat` on SATCOMP-2023 (d) to solve each test problem in seconds (for problems that are solved by at least one solver). Each problem is represented by a dot whose location indicates the solving time of each method. The dots on the dashed lines at 5,000 seconds indicate failures. It is evident that more dots are present in the lower triangular areas, indicating that there are more problems on which `NeuroBack-Kissat` outperforms both `Default-Kissat` and `Random-Kissat`.

sat versus unsat problems underpins the varying degrees of effectiveness observed in NeuroBack's performance.

## 10.3 SETTING UP THE MEMORY LIMIT FOR NEUROBACK-KISSAT

In our experimental setup, which includes a machine equipped with 256GB of memory running 64 solver instances in parallel, we have conservatively set the SAT formula size threshold at 135 MB. This ensures that the memory usage of each solver instance does not exceed our specified memory threshold of 10GB. This threshold setting is based on our practical experience. Increasing this threshold could potentially lead to memory contention issues. Users might choose to adjust the formula size threshold based on their machine's memory capacity. Alternatively, they might simply establish a memory threshold for each solver instance based on their machine's memory capacity and allow model inference to proceed until this threshold is reached, which typically incurs an overhead of no more than a few seconds.

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
