# OpenReview forum: "NeuroBack: Improving CDCL SAT Solving using Graph Neural Networks"
_ICLR.cc/2024/Conference — ICLR 2024 poster_

### Official Review · Reviewer_bkSL · 2023-10-29

**Soundness:** 3 good
**Presentation:** 3 good
**Contribution:** 2 fair
**Rating:** 5
**Confidence:** 4

**Summary:**

This paper studies the proposition satisfiability (SAT) problem, which is an NP-complete problem. Recent algorithms focus on enhancing CDCL, the mainstream algorithm for solving SAT problems, using Graph Neural Networks (GNNs). While this can make the  solving process more effective, these methods require online model inferences, which consumes substantial GPU resources. In this paper, the authors design an approach called NeuroBack, which uses the trained model and can be executed on the CPU, avoiding the dependence on GPU resources. The authors claim that enhancing the state-of-the-art SAT solver Kissat with NeuroBack can achieve better results than Kissat itself.

**Strengths:**

1. NeuroBack gets rid of the GPU resource dependence of the solving process and improves the practicality of using GNN to enhance SAT solving.

2. The authors conduct experiments to evaluate the performance of their proposed method.

**Weaknesses:**

I have several comments regarding the experiment part, which are listed below.

1. Regarding the baseline competitor. As described in the experiment part, the baseline competitor is kissat. Although kissat can be regarded as the latest breakthrough in the community of SAT solving, lots of kissat’s variants have been proposed since the introduction of kissat (as can be observed in the recent editions of SAT competitions). However, the authors only compare their solver with vanilla kissat. As a submission to a top-tier conference, this is not that thorough.

2. Regarding the evaluation on SAT Competition 2022’s datasets. Actually, as a submission in SAT solving, the tradition is to evaluate their proposed solver on the dataset from the latest edition of SAT Competition. In fact, as of the submission deadline of ICLR, the latest edition of SAT Competition is SAT Competition 2023. According to the official website of SAT Competition 2023 (https://satcompetition.github.io/2023/), the dataset of SAT Competition 2023 was published in July, 2023. Since the submission deadline of ICLR is due on September 28. 2023, there left two months for the authors to conduct the comparative experiments on the dataset of SAT Competition 2023. Hence, such lack of experiments is indeed a minus.

**Questions:**

Please see my comments listed in "Weaknesses".

**Details Of Ethics Concerns:**

Not applicable.

---

> ### Author Response · Authors · 2023-11-20
> **Response to Reviewer bkSL**
>
> Thank you for your valuable insights and suggestions. We hope to address your concerns regarding our paper.
>
> **Concern 1. Try Kissat variants**
>
> As the latest significant breakthrough in SAT solving, Kissat represents a strong and widely recognized standard in the community.  By comparing our solver to the most recent well-established SAT solver, we aimed to provide a clear and straightforward evaluation of NeuroBack's capabilities. We acknowledge that numerous variants of Kissat have emerged since its introduction, many of which have demonstrated notable performance in recent SAT competitions. However, these variants often include specific optimizations or modifications tailored to competition settings, which may not generalize well across broader settings. Our objective was to evaluate the effectiveness of our neural approach in a more generalized context, which made comparing it with the vanilla Kissat more suitable than conducting an exhaustive comparison with each Kissat variant. Furthermore, we examined the top three Kissat variants in the SAT competitions over the last three years, and found that all of their optimization techniques are orthogonal to NeuroBack's phase initialization technique. Given NeuroBack's already demonstrated efficiency gains with vanilla Kissat, we anticipate it will also positively impact these Kissat variants. Our future research efforts will focus on empirically validating this hypothesis.
>
>
> **Concern 2. Try SATCOMP-2023**
>
> Thank you for pointing this out. We have evaluated NeuroBack on SATCOMP-2023. Detailed results and analysis are now included and can be seen as supplementary material to the paper. In sum, NeuroBack-Kissat performs better than the baseline solver Kissat in both SATCOMP-2022 and SATCOMP-2023. Specifically, on SATCOMP-2023 problems, Kissat solved 198 problems and NeuroBack-Kissat 204, providing an improvement of 3.0%.
>
> In addition, since the default setting of Kissat is to simply initialize the phase of each variable as true, to further study NeuroBack effectiveness, we implemented an additional baseline solver, named Random-Kissat, which initializes the phase of each variable as either true or false at random. Detailed results and analysis are shown in the supplementary material of the paper. In summary, Random-Kissat performs better than Kissat in SATCOMP-2022, but worse in SATCOMP-2023; meanwhile, NeuroBack-Kissat performs better than Random-Kissat in both SATCOMP-2022 and SATCOMP-2023. More specifically, in SATCOMP-2022 problems, Random-Kissat solved 197 problems, whereas NeuroBack-Kissat solved 203, marking a 3.0% improvement. In SATCOMP-2023 problems, Random-Kissat solved 190 problems and NeuroBack-Kissat solved 204, demonstrating an improvement of 7.4%.
>
> Overall, these results suggest that the phase initialization provided by NeuroBack outperforms both the default and random phase initializations in Kissat, showing more effectiveness in solving SATCOMP-2022 and SATCOMP-2023 problems.

---

> > ### Comment · Reviewer_bkSL · 2023-11-22
> >
> > Thank you for the reply!
> >
> > In your reply, when solving the instances of SAT Competition 2023, you mention that Random-Kissat solves 190 instances and NeuroBack-Kissat solves 204 instances. However, according to the official results of SAT Competition 2023 (https://satcompetition.github.io/2023/results.html), the winner (SBVA Cadical, a variant of CaDiCaL solver) and the second-best solver (KissatMabProp PrNosym, a variant of Kissat solver) is able to solve 284 and 272 instances. Hence, there is a significant performance gap between your proposed method and the current state of the art. Since this is a paper submitted to a top-tier conference, I have to keep my score unchanged.

---

> > > ### Author Response · Authors · 2023-11-22
> > > **Further response to Reviewer bkSL**
> > >
> > > Thank you very much for your prompt response!
> > >
> > > There appears to be some confusion regarding our results on SATCOMP-2023. It's important to clarify that, out of the 400 problems presented in SATCOMP-2023, 353 were successfully inferred using only CPU resources and within the memory limit. Consequently, we applied NeuroBack-Kissat, Random-Kissat, and Kissat to these 353 inferred instances, rather than the full set of 400 instances. For more details, please refer to  the supplementary material of the paper. Additionally, it's crucial to acknowledge that our results should not be directly compared with those of the competition, as the experiments in each case were conducted using different computational resources. It's important to note that variations in computational resources can significantly influence the efficiency of SAT solving.

---

> ### Author Response · Authors · 2023-11-22
> **We eagerly anticipate your additional comments or questions**
>
> Dear Reviewer bkSL,
>
> Thank you once again for your valuable feedback! As the rebuttal phase is approaching, we eagerly anticipate your additional comments or questions. We hope our responses have met your concerns and would be grateful if you could consider an increase in your scores. Should you have further feedback, we are ready to actively incorporate it into our submission.
>
>
> Warm Regards,
>
> NeuroBack Authors

---

### Official Review · Reviewer_7KxS · 2023-10-29

**Soundness:** 3 good
**Presentation:** 3 good
**Contribution:** 4 excellent
**Rating:** 8
**Confidence:** 3

**Summary:**

This paper addresses the challenge of improving propositional satisfiability (SAT) problem-solving, a significant task in various research domains like planning, verification, and security. It introduces a novel approach called NeuroBack, which enhances Conflict-Driven Clause Learning (CDCL) SAT solvers using Graph Neural Networks (GNNs. The innovation lies in two key insights: predicting variable phases that frequently appear in satisfying assignments and requiring just one query to the neural model before starting the SAT solving process. Once trained, NeuroBack can operate entirely on the CPU, reducing the reliance on GPU resources.

The authors developed a new dataset, DataBack, comprising 120,286 data samples, to train NeuroBack and implemented it as an enhancement to the state-of-the-art SAT solver, Kissat. By incorporating NeuroBack into Kissat, the SAT solver exhibited a 5.2% improvement in problem-solving effectiveness, as demonstrated in the SATCOMP-2022 competition dataset. It also improved solving efficiency, resulting in an average time saving of 117 seconds per problem. This research introduces the first practical neural approach to enhance CDCL SAT solving without requiring GPU resources, provides a new dataset, and offers public access to the NeuroBack model and NeuroBack-Kissat source code.

**Strengths:**

Providing a dataset
Competing with SAT 2022

**Weaknesses:**

I think some citations are missing

https://cs.stanford.edu/~jure/pubs/g2sat-neurips19.pdf
Neurogift: Using a machine learning-based sat solver for cryptanalysis
Role of Machine Learning for Solving Satisfiability Problems and its Applications in Cryptanalysis

It would be nice to have a use case in Cryptanalysis (solving AES or small AES instance)

**Questions:**

-

**Details Of Ethics Concerns:**

-

---

> ### Author Response · Authors · 2023-11-20
> **Response to Reviewer 7KxS**
>
> We sincerely appreciate your positive feedback and constructive comments. Your specific and insightful comments are invaluable for enhancing the quality of our paper. We will definitely incorporate the recommended citations in the next version of our paper. Additionally, your suggestion to include a use case study in Cryptanalysis is truly stimulating. We are currently gathering benchmarks in Cryptanalysis and look forward to evaluating our NeuroBack in this context. Your guidance has been instrumental in refining our work.

---

### Official Review · Reviewer_HXv9 · 2023-11-01

**Soundness:** 3 good
**Presentation:** 3 good
**Contribution:** 2 fair
**Rating:** 5
**Confidence:** 4

**Summary:**

This paper is devoted to improving propositional satisfiability (SAT)
solving by means of utilizing modern machine learning (ML) technology.
Namely, the paper proposes to make use of a GNN architecture for
representing CNF formulas, which is trained to determine the variable
polarity / phase following the ideas behind backbone literals.
Backbone literals of a CNF formula are literals that must necessarily
be satisfied by every satisfying assignment of that formula. The
authors argue that the GNN architecture they propose can efficiently
determine the polarities of all variables: including those appearing
in backbone literals but also any other variables in the formula.
Afterwards, when the predicted polarities are obtained, a
state-of-the-art SAT solver can be bootstrapped with these variable
phases with the hope that such initialization can boost the solver's
performance. Experimental results shown in the paper demonstrate the
effectiveness of this idea if implemented on top of a modern SAT
solver called Kissat.

**Strengths:**

- The paper is clearly written and easy to follow. Normally, papers on
  applying ML for improving combinatorial problem solving are written
  in Greek, if seen from the perspective of a researcher with
  expertise in the combinatorial problem of interest. This paper
  serves as a nice exception.

- The idea is reasonable. It is simple to implement and it can be used
  with any SAT solver.

- The experimental results reported although not amazing look solid.

**Weaknesses:**

- Although everything the paper describes is described well, there are
  bits that aren't detailed sufficiently - some ML people may find it
  to be a minus.

- The paper says nothing about the usability of this heuristic for
  unsatisfiable instances. There are no backbones in those but I
  presume some variable phases may still be more useful in practice
  than the others.

- Despite the claimed experimental results, nothing is shown for the
  unsatisfiable instances. This and the point above can be joined
  together.

- Although Neuroback+Kissat solves 10 more instances out of 308, there
  are 92 more (308 + 92 = 400) where Neuroback fails to do anything
  useful. Hence, the proposed solver configuration must clearly lose
  to Kissat on those 92 instances as it spends additioanl time with no
  effect. If this understanding of mine is correct then the results
  aren't so positive.

- Minor1: in CDCL description, the algorithm undoes not only
  *wrong decisions* but also propagated literals.

- Minor2: in CDCL description, the conventional algorithm backtracks
  to the *latest* decision level where the conflict is resolved - not
  *earliest*.

**Questions:**

- Can you comment on the use of this heuristic for unsatisfiable
  instances? Have you tried this? If yes, what is the performance
  compared to Kissat? If not, why?

- Can you comment on losing on 82 instances to Kissat if we consider
  all the 400 instances in the benchmark set? Am I missing anything?

---

> ### Author Response · Authors · 2023-11-20
> **Response to Reviewer HXv9**
>
> Thank you for the detailed comments and questions. In the following, we hope to address the stated weaknesses of our paper.
>
> **Concern 1. Comment on the use of NeuroBack on unsat instances**
>
> Thank you for bringing this up! In our experiments, we did not filter the benchmarks based on their satisfiability. Therefore, the 308 instances certainly contain unsatisfiable instances. Upon detailed analysis, we discovered that there are 106 unsatisfiable instances and 99 satisfiable instances that are solved by either Kissat or NeuroBack-Kissat.
>
> For the 106 solved unsatisfiable instances, NeuroBack-Kissat outperformed Kissat in 63 cases (59.4%) while Kissat outperformed NeuroBack-Kissat in only 35 instances (33.0%).  For the 99 solved satisfiable instances, NeuroBack-Kissat outperformed Kissat in 57 instances (57.6%), while Kissat outperformed NeuroBack-Kissat in 40 instances (40.4%). Interestingly, our analysis revealed that while NeuroBack-Kissat showed a slightly higher improvement rate in unsatisfiable instances compared to satisfiable ones (59.4% vs 57.6%), the extent of improvement was more significant in satisfiable instances. On average, NeuroBack-Kissat enhanced the performance of satisfiable instances by 62.4%, compared to an average improvement of only 9.0% in unsatisfiable instances.
>
> The experimental results demonstrate that NeuroBack's predicted variable phases can enhance the efficiency in solving unsatisfiable instances. However, as expected, NeuroBack shows a more pronounced improvement in solving satisfiable instances. We will add these results in the next version of our paper.
>
> **Concern 2. Comment on the rest instances comparing to Kissat**
>
> Given that the NeuroBack model has a relatively small size, the primary memory consumption during model inference stems from the size of the graph representations of SAT formulas. To effectively manage memory contention, we have adopted a straightforward yet efficient approach: if the size of a SAT formula surpasses a predetermined size threshold, we default to using Kissat for solving the formula. This size checking is very quick, typically taking a few milliseconds. Therefore, the overhead for this process in the remaining 82 instances is minimal and can be considered negligible.

---

> ### Comment · Reviewer_HXv9 · 2023-11-21
> **Reply to authors**
>
> Thank you for the response.
>
> #### **Concern 1.**
>
> I believe the paper should provide a reader with a discussion on how the approach helps solving unsatisfiable instances too, as well as why it does so. I hope you agree that it is something not entirely intuitive because the approach targets detecting backbone phases, which aren't present in unsatisfiable instances at all. The discussion must also comment on the performance on SAT and UNSAT instances and differentiate the two.
>
> #### **Concern 2.**
>
> In that case, the paper should clearly detail the experimental setup saying what thresholds were used (if any) and how its assessment affects the performance (even it spends literally no time, the paper should say it). In either case, what is the threshold and what happens if you raise it? Does it significantly deteriorates the performance? This should be discussed in the paper because this seems to be a clear practical limitation of your approach.

---

> > ### Author Response · Authors · 2023-11-22
> > **We eagerly anticipate your additional comments or questions.**
> >
> > Dear Reviewer HXv9,
> >
> > Thank you once again for your valuable feedback! As the rebuttal phase is approaching, we eagerly anticipate your additional comments or questions. We hope our responses have met your concerns and would be grateful if you could consider an increase in your scores. Should you have further feedback, we are ready to actively incorporate it into our submission.
> >
> > Warm Regards,
> >
> > NeuroBack Authors

---

> > ### Author Response · Authors · 2023-11-23
> > **Updated appendix in the supplementary material to address the concerns**
> >
> > Thank you once more for your valuable insights! We have revised the appendix in the supplementary material to address Concern 1 in Section 2 and Concern 2 in Section 3, respectively.  We hope our responses have met your concerns and would be grateful if you could consider an increase in your scores. Should you have further feedback, we are ready to actively incorporate it into our submission.

---

> ### Author Response · Authors · 2023-11-22
> **Further response to Reviewer HXv9**
>
> Thank you for your insightful guidance! We will add the following clarifications in the next version of our paper. Please let us know if you have any further suggestions.
>
> **Further Response to Concern 1.**
>
> 1) a discussion on how and why the approach helps solving unsatisfiable instances
>
> In fact, NeuroBack predicts phases of all variables appearing in the majority of the satisfying assignments, not only the backbone phases. We certainly agree that how the predicted phases help in solving unsatisfiable instances is not entirely intuitive. We really appreciate your insightful question.  Here's our clarification: For unsatisfiable instances, NeuroBack's phase predictions can aid in directing the search towards the unsatisfiable part of the search space. While NeuroBack cannot satisfy all components of a given SAT problem, it may predict phases that satisfy certain components, thereby allowing the solver to concentrate on the unsatisfiable part. Furthermore, in modern SAT solvers like Kissat [1], an assignment that falsifies the fewer clauses is often preferred in the searching loop, allowing the solver to specifically target the unsatisfiable portions of the clause set. Consequently, the phases predicted by NeuroBack can facilitate identifying an assignment that reduces clause falsification, thereby enhancing solving unsatisfiable instances.
>
> 2) comment on the performance on SAT and UNSAT instances and differentiate the two
>
> The results shown in the previous comment indicate that NeuroBack achieves more notable success in solving satisfiable instances compared to unsatisfiable ones. This distinction stems from the inherent nature of these problems. In satisfiable instances, a complete satisfying assignment exists, where each variable is assigned a phase leading to a solution. Conversely, in unsatisfiable instances, only partial satisfying assignments exist, with phases assigned to just a subset of variables. Consequently, the phases predicted by NeuroBack have a generally greater impact in resolving satisfiable instances. This is because, for these instances, the predicted phases can contribute directly to finding a satisfying assignment. In contrast, for unsatisfiable instances, the utility of predicted phases is somewhat restricted to identifying partial solutions or refining the search scope. This fundamental difference in the nature of satisfiable versus unsatisfiable problems underpins the varying degrees of effectiveness observed in NeuroBack's performance.
>
>
>
> **Further Response to Concern 2.**
>
> Thank you for pointing this out! In our experimental setup, which includes a machine equipped with 256GB of memory running 64 solver instances in parallel, we have conservatively set the SAT formula size threshold at 135 MB. This ensures that the memory usage of each solver instance does not exceed our specified memory threshold of 10GB. This threshold setting is based on our practical experience. Increasing this threshold could potentially lead to memory contention issues. Users might choose to adjust the formula size threshold based on their machine’s memory capacity. Alternatively, they might simply establish a memory threshold for each solver instance based on their machine's memory capacity and allow model inference to proceed until this threshold is reached, which typically incurs an overhead of no more than a few seconds. We will add this clarification to the next version of our paper.
>
> **References**
>
> [1] Biere, Armin, and Mathias Fleury. "Chasing target phases." Workshop on the Pragmatics of SAT. 2020.

---

### Official Review · Reviewer_w5iE · 2023-11-01

**Soundness:** 3 good
**Presentation:** 3 good
**Contribution:** 3 good
**Rating:** 6
**Confidence:** 4

**Summary:**

The NeuroBack paper proposes a new approach to improve CDCL SAT solving using Graph Neural Networks. The proposed approach, called NeuroBack, gives an initial assignment to all variables and queries the neural model only once, allowing it to execute exclusively on the CPU and making GNN improvements practical. The paper also introduces a new dataset called DataBack and implements NeuroBack as an enhancement to a state-of-the-art SAT solver called Kissat. The authors evaluate NeuroBack on a variety of benchmarks and show that it outperforms Kissat and other state-of-the-art solvers on many instances.

**Strengths:**

Using ML to enhance CDCL SAT solver is promising research, which is more likely to yield improvement in SAT solving than end-to-end learning on SAT. I like the idea in this paper of using ML to give an initial assignment for the CDCL solvers by training on predicting the value of backdoor variables. The paper mentions the importance of backdoor valuables and the intuition of why training on backdoor variables can help predict the values for all variables that appear in the majority of the variables.

It is also promising to see that NeuroBack is actually able to improve Kissat, as state-of-the-art SAT solvers are well-engineered and very hard to optimize.

**Weaknesses:**

While training on backdoor variables yields a good predictor for the value of all variables, why this approach works is still a bit mysterious. It would be helpful to have more experiments showing that initialization is actually better than a random assignment. Therefore I would like to see some behavior studies of NeuroBack on different kinds of benchmarks. Possible ways include comparing the distance from this initial assignment given by NeuroBack with the nearest solution. Or evaluating the prediction accuracy of the value of a specific variable given by NeuroBack with the majority value of this variable in all solutions (this requires listing all solutions of a formula, which is only doable on small formulas).

The experiment section also lacks a comparison with other initialization methods for SAT solving, as Neuro-back is in essence an initialization method.

I am not quite convinced by the claimed advantage that NeuroBack only needs to be called once. This is a trivial property for any initialization approach for SAT solving. I believe that a more interactive collaboration between GNN and SAT solvers can further improve the performance of STA solvers, even if GNN needs to be called multiple times (correct me if I am wrong). I would be happy to change the evaluation if my concerns can be addressed.

Minor comments:
I would use "value" instead of "phase" and "initial assignment" instead of "initialization" to make it clear.

**Questions:**

See the above comments.

---

> ### Author Response · Authors · 2023-11-20
> **Response to Reviewer w5iE**
>
> Thank you for the thoughtful and in-depth comments. We hope to address your concerns regarding our paper.
>
> **Concern 1: More experiments comparing neural initialization to random assignment**
>
> To address the reviewer's concern, we implemented a random initialization solver based on Kissat, named Random-Kissat, which initializes the phase of each variable to true or false randomly. We also included benchmarks from the most recent SAT competition, SATCOMP-2023, as "different kinds of benchmarks'' requested by the reviewer. Detailed results and analysis are now included and can be viewed in the supplementary material of the paper.
>
> In summary, Random-Kissat performs better than Kissat in SATCOMP-2022, but worse in SATCOMP-2023; meanwhile, NeuroBack-Kissat performs better than Random-Kissat in both SATCOMP-2022 and SATCOMP-2023. More specifically, in SATCOMP-2022 problems, Random-Kissat solved 197 problems, whereas NeuroBack-Kissat solved 203, making a 3.0% improvement. In SATCOMP-2023 problems, Random-Kissat solved 190 problems and NeuroBack-Kissat solved 204, demonstrating an improvement of 7.4%. Overall, these results indicate that the phase initialization provided by NeuroBack is more effective than random assignment in both SATCOMP-2022 and SATCOMP-2023 problems.
>
> **Concern 2: not convinced by the claim that NeuroBack only needs to be called once**
>
> We partially agree with the reviewer about the benefits of interactive collaboration between GNN and SAT solvers, where the GNN refines its predictions based on the current state of SAT solving, potentially improving neural prediction accuracy. However, it is crucial to consider the potential performance bottleneck arising from such interactions. The increased demand for GPU resources in parallel SAT solving, a prevalent approach in industrial applications for tackling complex problems, could significantly slow down the overall solving process. As more solver runs are executed in parallel, the competition for GPU resources intensifies. Lacking GPU resources may become a major bottleneck in achieving the expected parallel performance. Notably, if the GNN already demonstrates high prediction accuracy before the start of SAT solving, the frequent use of GPU resources for minor adjustments in predictions through online inferences would be less efficient than relying on the initial offline predictions.
>
> The reviewer's concern arises from the application of NeuroBack's phase prediction.
> As outlined in the discussion and future work sections, our current approach only focuses on applying NeuroBack's phase predictions to initialize the variable phases in SAT solvers. However, there exists various approaches for applying NeuroBack predictions, either statically or dynamically, each potentially enhancing the SAT solving efficiency. For instance, NeuroBack phase predictions could be leveraged dynamically during rephasing to adjust or diversify the exploration of the search space. In future research, we aim to delve deeper into the broader applications of NeuroBack predictions, exploiting NeuroBack's full potential in SAT solving.

---

> > ### Comment · Reviewer_w5iE · 2023-11-22
> >
> > Thank you for the response and the results, which do address some of my concerns. I will be happy to increase my score.

---

> > > ### Author Response · Authors · 2023-11-23
> > > **Thank you!**
> > >
> > > Thank you for raising your evaluation score. We truly appreciate it! If you have any additional concerns, we are prepared to actively address and incorporate them into our submission.

---

> ### Author Response · Authors · 2023-11-22
> **We eagerly anticipate your additional comments or questions.**
>
> Dear Reviewer w5iE,
>
> Thank you once again for your valuable feedback! As the rebuttal phase is approaching, we eagerly anticipate your additional comments or questions. We hope our responses have met your concerns and would be grateful if you could consider an increase in your scores. Should you have further feedback, we are ready to actively incorporate it into our submission.
>
> Warm Regards,
>
> NeuroBack Authors

---

### Author Response · Authors · 2023-11-20
**Response to All Reviewers**

We would like to extend our sincere gratitude to all reviewers for your insightful and constructive feedback. We deeply appreciate the time and effort you have invested in reviewing our work, and we are grateful for the opportunity to refine our paper based on your expert guidance. Our responses to each of your comments are detailed below. We earnestly hope that our rebuttal effectively addresses your concerns. If you find our revisions satisfactory, we would be grateful if you could consider increasing your evaluation score. Should there be any remaining issues, please do not hesitate to share them. We are committed to continuously addressing your comments and refining our submission.

---

### Author Response · Authors · 2023-11-22
**We eagerly await any further comments or questions you might have.**

Dear Reviewers,

Thank you once again for your insightful and constructive feedback, which have been extremely helpful in enhancing our work. We apologize for the slight delay in our response, as we were engaged in additional experiments to comprehensively address your comments. With the rebuttal phase imminent (due on November 22 AOE), we eagerly await any further comments or questions you might have.

Warm regards,

The Authors

---

### Author Response · Authors · 2023-11-23
**The main paper and the supplementary material have been updated**

Dear Reviewers,

We sincerely appreciate your insightful feedback! Following your suggestions, we have incorporated our discussions into both the main paper and the supplementary material. Your guidance has been instrumental in enhancing the quality of our work, and we are deeply grateful for your assistance.

Thanks,

NeuroBack Authors

---

### Meta-Review · Area_Chair_VqLp · 2023-12-06

**Metareview:**

This work proposes a new approach to improve CDCL SAT solving using Graph Neural Networks. The authors also developed a new dataset, named DataBack, to train NeuroBack and implemented it as an enhancement to the state-of-the-art SAT solver, named Kissat. Overall, the reviewers acknowledge the technical soundness and rich experimental results. The main concerns lie in lacks of detailed comparison with other initialization methods for SAT solving, results for unsatisfiable instances, and the performance on the dataset of SAT Competition 2023. Considering that after the rebuttal, this paper receives two positive ratings (6 and 8) and two slightly negative ratings (5 and 5), I lean towards acceptance.

**Justification For Why Not Higher Score:**

Please see the meta-review

**Justification For Why Not Lower Score:**

Please see the meta-review

---

### Decision · Program_Chairs · 2024-01-16

Accept (poster)